# Soil Bacterial Community Structure in Turfy Swamp and Its Response to Highway Disturbance

**DOI:** 10.3390/ijerph17217822

**Published:** 2020-10-26

**Authors:** Yuanyuan He, Yan Xu, Yan Lv, Lei Nie, Hong Wang

**Affiliations:** 1College of Construction Engineering, Jilin University, Changchun 130026, China; hyy20@mails.jlu.edu.cn (Y.H.); xuyan8102@jlu.edu.cn (Y.X.); nielei@jlu.edu.cn (L.N.); 2Collge of Civil Engineering, GuiZhou University, Guiyang 550025, China; wanghong15@mails.jlu.edu.cn

**Keywords:** highway, turfy swamp, bacterial community, road drainage, environmental factors

## Abstract

In recent years, the construction and development of highways in turfy swamp areas has been very common. When highways pass through turfy swamps, they can change the local soil, vegetation and hydrological environment, but the impact on soil microorganisms is unclear. We studied the impact of highways on soil microbial communities and diversity in three turfy swamps. Soil samples were collected in the affected area (distance from the expressway 10 m) and control area (distance from the expressway 500–1000 m), and the soil properties, heavy metal content and microbial composition were measured. Subsequent statistical analysis showed that soil organic carbon (SOC), total nitrogen (TN), Cd, Cr, Zn, Cu, density and especially water table (WT) are the main driving forces affecting the composition of microorganisms. The WT and density can also be used to predict the change trend of the ratio of proteobacteria to acid bacteria, reflecting the soil nutrient status. In general, the composition of soil microorganisms in turfy swamp is mainly affected by road drainage and heavy metal emissions. This research provides new insights into the impact of highways on turfy swamps from the perspective of bacterial diversity and community composition, and it also provides a basis for the restoration of the wetland ecological environment.

## 1. Introduction

Wetland ecological environments, especially turfy swamps, are complex systems affected by the interactions of geological, hydrological, physicochemical and biological factors [1]; they provide habitats for biota on the earth and play a key role in global carbon cycles [2,3,4]. The high water level and subsequent anaerobic environment lead to the imbalance between primary productivity and microbial decomposition of organic matter, which is the important reason for the formation of the carbon sink function of turfy wetlands. However, because of the rapid development of highways in recent decades in China, many highways inevitably invade turfy wetlands [5,6], which has caused many negative impacts on the wetland environment, including soil erosion, water quality deterioration and vegetation destruction [7]. The sharp increasing of negative effects will lead to the imbalance of wetland self-regulation functions and even cause the functional transformation from “carbon sink” to “carbon source”. Microorganisms are important decomposers of soil organic matter in turfy wetlands, and their responses to environmental changes caused by highways are poorly understood.

In natural turfy swamps, the microbial assemblages are determined by vegetation type, soil nutrient status and degree of flooding, and they change with depth due to redox conditions and substrate availability change [8,9,10]. High microbial activity and diversity occur in nutrient-rich turfy wetlands. In addition, Proteobacteria and Acidobacteria are two main microbial phyla in turfy wetlands, and their relative abundance show the opposite trend when the environmental factors change [11]. Many studies have found that these natural gradients in microbial assemblages in turfy swamps can be disturbed by anthropogenic activities and local changes in environmental conditions. For example, Sun [12] found that the tree species and forest management have a strong impact on the bacterial diversity and community structure at a boreal peatland of Central Finland, and Urbanová and Bárta [13] found that the structure of soil bacteria and archaea community changed after the long-term drainage of a peat wetland. All these findings indicate that soil microbes are extremely sensitive to the interference from external environment caused by human activities in turfy swamps.

Heavy metal pollution is an important environmental problem caused by highway traffic, which has attracted many researchers [14,15,16,17]. These metals (chromium(Cr), cadmium(Cd), copper(Cu), zinc(Zn), lead(Pd)) are accumulated into the roadside environment through dry and wet deposition, and maintaining high concentrations of these pollutants in soils poses a threat to soil microorganisms due to their low degradation and high toxicity [18,19]. Turfy wetlands are also facing the same problem, as previously reported by Wang et al. [6]; their study indicated that traffic-related metals presented nonpollution to severe pollution levels. Zhao conducted heavy metal content testing and microbial gene sequencing in soil along the Qinghai-Tibet Highway and Qinghai-Tibet Railway. Their study reveals that heavy metal contamination from roads that have been open for more than a decade has affected soil bacterial abundance and bacterial community structure [20]. By correlation analysis and redundancy analysis, Zhang found that Cr and Cd were the major factors that influenced soil bacterial community changes in East Dongting Lake wetland, China [21]. However, drainage is another important problem affecting the ecological environment of turfy wetlands [12,22]. Turfy swamps are usually located in valleys with perennially accumulated water, and in order to ensure the safe operation of highways, drainage ditches parallel to the highway are built, resulting in a large amount of water loss. Many scholars have reported that wetland drainage can change the physical properties such as density and water content of the original soil [23,24] and cause soil nutrient loss [25]. In any case, there is still a lack of knowledge about how soil microorganisms respond to road drainage and heavy metal pollution in turfy swamps.

A better understanding of soil microbial assemblages after road drainage and heavy metal pollution can provide a theoretical foundation for the management and restoration of turfy swamps and can establish a basis for highway construction decisions in turfy wetlands. In this paper, we took soil samples along the turfy swamp highway and in the control area, tested their physical and chemical properties and measured their bacterial composition using high-throughput sequencing technology. Through comparative analysis of soil physical and chemical properties, heavy metal content and bacterial diversity in the affected area and the control area, it was found that water table (WT) is the main factor affecting the structure of soil bacterial community, and its changes are caused by road drainage. Heavy metal emissions caused by highway traffic will also affect the structure and composition of the bacterial community in the soil. The main purpose of this research is to investigate the composition of the bacterial community in the turfy swamp and the changes in the bacterial community structure caused by the impact of highways on its environment.

## 2. Materials and Methods

### 2.1. Site Description and Sample Collection

This study was performed in the Changbai Mountain area, Jilin Province, where there are a large number of turfy swamps. This area has a typical temperate continental monsoon climate with annual temperature and rainfall of 2–6 °C and 400–900 mm, respectively. The main vegetation of the turfy swamp in this area includes *Carex meyeriana*, *Thelypteris palustris* var. *pubescens* Fernald and *Sanguisorba tenuifolia* var. *alba*. Soil samples were collected from three similar and independent turfy swamps, namely Jiangyuan (JY; N43°7′, E128°1′), Longquan (LQ; N42°25′, E126°36′) and Huangsongdian (HSD; N43°39′, E127°39′) (Figure 1), which have been seriously disturbed by highway-related activities in recent years [6].

Soil sampling was performed in an area adjacent to the highway (<10 m; affected area) and in an area away from the highway (500–1000 m; control area) in July 2016. In each area, three quadrats (4 m × 4 m per quadrat) with 100 m spacing were arranged as triplicate sampling sites (Figure 1). The upper layer of soil (0–30 cm) was sampled using a corer in each quadrat (each sample including 5 subsamples mixed together). After carefully removing root debris, soil samples were divided into two parts: one was naturally air-dried through a 2 mm nylon sieve to determine physical and chemical properties; the other sample was stored in a cryogenic chamber at −80 °C for later throughput. In addition, a perforated PVC tube was set in each quadrat to monitor the water level during the growing season. The naming rules for soil samples in these three sites were as follows: JY affected area is JY1–JY3, and JY control area is CKJ1–CKJ3; LQ affected area is LQ1–LQ3, and LQ control area is CKL1–CKL3; HSD affected area is HSD1–HSD3, and HSD control area is CKH1–CKH3.

### 2.2. Physicochemical Properties Analyses

To determine the pH, soil organic carbon (SOC), total nitrogen (TN), total potassium (TK) and total phosphorus (TP) of the soil, the soil was dried naturally at room temperature. Coarse particles and grass roots were removed from the soil, and the soil was ground up and passed through a 2 mm nylon sieve for later use. Soil pH was measured using a pH meter (Model PHS-3C pH meter, INESA, Shanghai, China) at 1:2.5 (soil to water) after 30 min of shaking [26]. 

Soil organic carbon (SOC) was determined by the classical potassium dichromate oxidation–outer heating method according to standards of forestry (LY/T 1237-1999 and LY/T 1228-2015). The main steps are as follows: (1) Weigh 0.1000–0.5000 g (accurate to 0.0001 g) of soil sample into a rigid test tube and add 0.1 g of silver sulfate. (2) Add 5.00 mL of 0.8000 mol/L standard solution of potassium dichromate, and then inject 5.00 mL of sulfuric acid into the syringe and shake well. (3) Put the tubes in an oil bath pan at 170 to 180 °C and keep the solution in them boiling for 5 min. (4) Take out the test tube and let it cool down, then wash the solution into 250 mL conical flask, with the volume of the flask is controlled at 60–80 mL. (5) Add 3–4 drops of o-phenanthroline indicator, titrate the solution with 0.2 mol/L standard solution of ammonium ferrous sulfate to the end of the solution from orange-yellow by blue-green to brown-red.

Soil total nitrogen (TN) was determined using the Kjeldahl method according to standards of forestry (LY/T 1237-1999 and LY/T 1228-2015). The main steps are as follows: (1) Weigh 0.5000 g of soil sample, add it to a dry digestion tube, and add 1.5 g of reducing mixture catalyst. (2) Add 5 mL of concentrated sulfuric acid with a syringe and put it on a digester in a fume hood for 1.5 h until the contents are clear and light blue. (3) Place the triangular bottle under the socket of the condensation tube and submerge the mouth of the tube in boric acid solution (absorbent in triangular bottles: 20 mL of 2% boric acid). (4) When the receiving liquid turns blue after distillation for 5 min, leave the lower end of the condenser tube at the boric acid level, and then flush the outside of the tube with distilled water. (5) Titrate with 0.001 eq. of standard solution of hydrochloric acid until red and record the volume of the consumed standard solution. (An additional set of blanks is required, and the steps are identical except that no soil sample is added.)

Soil total phosphorus (TP) was measured according to Mo-Sb colorimetric method after digestion HF-HClO_4_-HNO_3_. HF-HClO_4_-HNO_3_ digestion steps: (1) Weigh 0.2–0.5 g (accurate to 0.0001 g) of soil sample and put it into a 50 mL PTFE digestion tube. (2) Rinse the adhered soil on the inner wall to the bottom of the tank carefully with a small amount of deionized water through the bottle washing nozzle. Place the digestion tube in the hole of the graphite block of the digester, add 10 mL of HNO_3_, boil at 100 °C and maintain at this temperature for 60 min, cool for 10 min. (3) Add 5 mL of hydrofluoric acid and 1 mL HCLO_4_ and continue to heat to 150 °C; heat for 150 min, then cool to room temperature. (4) Add 0.5 mL of nitric acid to dissolve the residue into a 50 mL volumetric jar. Wash the digestion tube several times with a small amount of deionized water and transfer the wash solution along with it to the volumetric bottle.

The main steps of TP determination are as follows: (1) Absorb 5 mL of solution accurately in a 25 mL volumetric bottle, add 2 mL of ammonium molybdate sulfate solution and 2 drops of antimony potassium tartrate solution, mix well and dilute to the scale line. (2) Add ascorbic acid solids, punch and mix well and leave for 5–20 min. (3) Measure the absorbance in a type 72 spectrophotometer at 680 nm wavelength with water as the reference. Then, find the phosphorus content in the standard curve.

Soil total potassium (TK) was analyzed using flame photometric method after digestion HF-HClO_4_-HNO_3_. The main steps are as follows: (1) Absorb 5–10 mL solution in 50 mL volumetric bottle, then fix the volume to scale with water. (2) Determine it on flame photometer and record the reading of flow detector. (3) Find the potassium concentration from the standard curve. Soil density was determined by the cutting ring method. The contents of Cr, Zn, Cu, Cd and Pb were extracted using the ICP-MAS (Q/JUTC010-2007) methods described by Wang [6]. 

### 2.3. DNA Extraction, PCR Amplification and Pyrosequencing

Genomic DNA was extracted from 250 mg of each soil sample using a PowerSoil DNA Isolation Kit (MoBio laboratories Inc., Carlsbad, CA, USA) as recommended by the manufacturer. The extracted DNA was purified using NanoDrop-1000 spectrophotometer (ThermoFisher, Waltham, MA, USA). The V4-V5 region of the bacterial 16S rRNA genes was PCR amplified using the primer sets of 515F (5′-GTGCCAGCMGCCGCGGTAA-3′) and 926R (5′-CCGTCAATTCMTTTGAGTTT-3′). The specific amplification process was as follows: 2 min at 94 °C; 25 cycles of 30 s at 94 °C for denaturation, 30 s at 56 °C for annealing and 30 s at 72 °C for extension; and the final extension at 72 °C for 5 min. In addition, the PCR amplification system included 10 μL of 5 × buffer, 1 μL of dNTP (10 mm), 1 U of Phusion DNA polymerase, 5–50 ng of template DNA, 1 μL of each F/R medial primer (10 mm) and ddH_2_O to a total volume of 50 μL. After PCR amplification, the PCR products were determined by analyzing 3 μL of product on 1.2% agarose gel. Next, the amplicons extracted from the 2% agarose gels were purified using the AxyPrep DNA Gel Extraction Kit (Axygen Biosciences, Union City, CA, USA) and quantified using FTC-3000TM real-time PCR (Funglyn Biotech Inc., Toronto, Canada), and the purified amplicons were pooled on an Illumina MiSeq platform (TinyGene Bio-Tech Co., Ltd., Shanghai, China) with the equimolar amounts and paired-end sequenced (2 × 300 bp) according to standard protocols.

### 2.4. Statistical and Bioinformatics Analysis

Raw pyrosequencing reads with an average quality score < 20 or the reads < 50 bp were trimmed off at 50 bp sliding window using Trimmomatic. Next, all paired reads with at least 20 base overlaps between forward and reverse reads were merged at a maximum mismatch ratio of 0.2 to form a chimeric sequence. To get more accurate results of bioinformatics analysis, the chimeric sequences need to be filtered by quality control, i.e., removing chimerism, ambiguous, homologous and singletons using Mothur software (v.1.39.5). Finally, the optimization sequences were compared with the RDP and database for species annotation, and the confidence threshold was set to 0.6. The operational taxonomic units (OTUs) with ≥ 97% similarity were clustered using Usearch (version 5.2.236). Based the OTUs, bacterial alpha diversity indices including Ace, Chao, Shannon and Simpson were calculated using Mothur. 

The one-way ANOVA and LSD tests were used to identify significant differences in the alpha diversity indices and the relative sequence abundances of bacterial phylum and genus between the affected area and control area (SPSS 21.0 software). The same approach was used for analyzing the significant differences in the environmental factors between the affected and control areas. In addition, correlation and multiple linear regression analyses were conducted using SPSS 21.0 software. Principal component analysis (PCA), redundancy analysis (RDA), ANOSIM test, variation partition analysis and heatmap analysis were executed using R vegan package (version 2.5-2), and surface fitting regression was performed using Origin 9.0 software. Monte Carlo permutation was used to test the significance of interpretation of environmental variables for species.

## 3. Results

### 3.1. Environmental Factors

Road drainage and pollutant emission had significant effects on the soil physicochemical properties, heavy metal concentrations and water table (WT) in turfy swamps. The soil organic carbon (SOC) and total nitrogen (TN) contents in control area of the three sites were significantly higher than those in the area affected by highway (Table 1). The soil total potassium (TK) and total phosphorus (TP) varied from 5.61 to 8.81 g kg^−1^ and 1.15 to 1.37 g kg^−1^, respectively, and there was no significant difference between the affected and control areas (*p* > 0.05), but TK showed a slight increase in the control area compared with the affected area. The chromium, cadmium, copper, zinc and lead (Cr, Zn, Cu, Cd and Pb) contents were in the ranges of 46.77–121.44 mg kg^−1^, 64.63–170.24 mg kg^−1^, 15.70–51.86 mg kg^−1^, 0.138–0.837 mg kg^−1^ and 17.56–22.45 mg kg^−1^, respectively. The concentrations of most determined metals, excepted for Pb, showed significant difference between the affected and control areas in turfy swamps. The WT ranged between −44.5 and 10.1 cm, and there was a significant difference between the affected and control areas in these three turfy swamps. The range of soil pH was rather narrow (4.82–5.88), with no significant difference between the affected and control areas in turfy swamp. In sum, the values of soil SOC and TN in affected area were significantly lower than those in the control area. For the metals, increased contents were found in the affected area due to traffic pollution.

### 3.2. Bacterial Diversity

A total of 698,532 high-quality reads with an average length of 412 bp were obtained from 18 samples after trimming and quality filtering at the three turfy swamps. In addition, the total number of OTUs obtained from the soil samples in the control and affected areas were 8638 and 7016, respectively, and the species coverage reached 98%. Based on 3% genetic distance, the rarefaction curves tended to approach the saturation plateau, indicating that the sequencing depth was sufficient and the majority of the bacterial diversity was included (Figure 2b). Species accumulation curves revealed that sample size was adequate for subsequent data analysis (Figure 2c).

The Ace, Chao, Shannon and Simpson indices were calculated to evaluate the abundance and diversity of soil bacterial communities, and the results are listed in Table 2. At affected areas, the Ace and Chao indices at JY and LQ sites were significantly higher than those at HSD site (*p* < 0.05). For the control area, the values of Ace and Chao were highest in JY, second in LQ and lowest in HSD, and there were significant differences between the three sampling sites (*p* < 0.05). In addition, it is worth noting that the Ace and Chao indices for three turfy swamps revealed similar trends, and their values at control area were significantly higher than those at the affected area (*p* < 0.05). For the Shannon and Simpson indices, however, there was no significant difference between sampling sites (*p* > 0.05). 

The results of the Pearson correlation analysis revealed that the main environmental factors examined, especially the SOC, TN, Cr, Zn, Cu, Cd, WT and the soil density, were significantly associated with the bacterial alpha-diversity indicators (Figure 2a). The SOC and TN showed a significant positive correlation with OTU and Chao index, and TN and Ace were also significantly positively correlated with Shannon index. Cr had a strong negative correlation with Simpson index, and Zn, Cu and Cd exhibited a significant negative correlation with Shannon and Simpson indices. Additionally, the soil density was significantly negatively correlated with OTU, Chao, Ace, Shannon and Simpson (*p* < 0.05), and WT was significantly positively correlated with OTU, Chao, Ace and Shannon (*p* < 0.001). The results of multiple linear regression analysis showed that only WT was positively correlated with microbial diversity, particularly with OTU (R2 = 0.652, *p* < 0.05), Chao (R2 = 0.561, *p* < 0.05), Ace (R2 = 0.531 *p* < 0.05) and Shannon (R2 = 0.524, *p* < 0.05) (Appendix A, Figure 3). 

### 3.3. Bacterial Community Composition

The relative abundance of bacterial community at the phylum level is shown in Figure 4. In total, 24 phyla were identified, and the predominant phyla at all the soil samples were Proteobacteria (28.53–40.28%), Acidobacteria (9.02–21.90%), Bacteroidetes (10.08–14.44%), Chloroflexi (6.65–8.49%), Planctomycetes (3.29–4.96%) and Ignavibacteriae (1.96–6.18%), with the remaining phyla accounting for less than 5% of the total. Proteobacteria was the most abundant phylum, and its relative abundance was significantly lower at the affected sites, while that of Acidobacteria was significantly higher (Table 3, *p* < 0.05). All other phyla were not significantly different, except for Omnitrophica which was significantly higher and Gemmatimonadetes which was significantly lower at the affected sites (Table 3).

At the genera level, the predominant genera were *Geobacter*, *Opitutus*, *Candidatus Solibacter*, *Syntrophus*, *Escherichia*, *Sideroxydans*, *Bryobacter* and *Rhizomicrobium* with an average relative abundance of 3.19%, 2.82%, 1.95%, 1.09%, 1.41%, 0.24%, 0.92% and 0.52%, respectively. The relative abundance of several genera differed significantly between the affected and control areas in the three turfy swamps (Table 3 and Table 4). The areas affected by highway had significantly (*p* < 0.05) higher relative abundance of *Rhizomicrobium* (0.92% A, 0.13% C), *Candidatus Solibacter* (2.61% A, 1.30% C), *Terrimonas* (0.61% A, 0.11% C), *Nitrospira* (0.58% A, 0.12% C) and *Gemmatimonas* (0.86% A, 0.05% C) than the control area, whereas *Geobacter* (4.77% A, 1.62% C), *Syntrophus* (0.07% A, 2.12% C), *Escherichia* (0.36% A, 2.47% C), *Syntrophorhabdus* (0.58% A, 0.12% C), *Smithella* (0.58% A, 0.12% C), *Syntrophobacter* (0.04% A, 0.86% C), *Methylobacter* (0.42% A, 0.14% C), *Longilinea* (0.04% A, 1.01% C) and *Leptolinea* (0.06% A, 0.30% C) were more abundant in control area soils, as shown in Table 4.

### 3.4. Relative Influences of Environmental Factors on Bacterial Community

Principal coordinates analysis (PCoA) of the bacterial communities was applied to reveal the distance and differences between the samples (Figure 5). The results showed that the soil samples obtained at the affected area were away from points of the control area in the three turfy swamps. The result of cluster analysis on genus level revealed that the 16S rRNA gene community compositions of the soils in the affected area significantly differed from those in the control area (Table 3). The results of ANOSIM testing performed on a Bray–Curtis matrix indicated that bacterial communities in affected areas were significantly different from those of the control areas in these three turfy swamps (phylum: R = 0.9538, *p* = 0.001; genus: R = 0.9609, *p* = 0.001), which may be due to the influence of environmental factors.

Redundancy analysis (RDA) clearly demonstrated the relationship between the bacterial community structure and environmental factors (Figure 6). The results indicated that all the twelve parameters explained 67.4% of the variation in the species data. Subsequent variance partitioning analysis revealed that the WT and physical and chemical properties, the five heavy metals and the interaction effects of these factors explained 20.66%, 15.29% and 44.77% of the total variation, respectively. Moreover, soil density, WT, Cd, Zn, Cu, Cr, TN, SOC and TK explained 41.9%, 41.3%, 33.5%, 31.4%, 22.3%, 21.0%, 18.0%, 16.0% and 14.4% of variance, respectively. Other factors, such as Pb, TP and pH only explained 9.3%, 2.8%, and 2.3% of total variance respectively. Ye et al. [27] have reported that Pr is a parameter for describing the significant level of correlation between individual environmental factors and microbial communities in RDA analysis. The Pr values were 0.001 for soil density, WT, Cd and Zn; 0.008 for Cu; 0.005 for Cr; 0.034 for TN; 0.030 for SOC; 0.043 for TK; and >0.05 for Pb, TP and pH in this study (Monte Carlo permutation test, 499 permutations). These results indicated that density, WT, Cd, Zn, Cu, Cr, TN, SOC and TK were the key factors affecting the microbial community.

Pearson’s correlation coefficients were calculated to further reveal the correlations between these environmental factors and the bacterial abundance (Figure 7). The abundance of *Geobacter* and *Methylobacter* were significantly and positively correlated with SOC, TN and WT (*p* < 0.01) and negatively correlated with density (*p* < 0.05). The abundance of *Syntrophus*, *Syntrophorh abdus* and *Syntrophobacter* of the phylum Proteobacteria and *Longilinea* of the phylum Chloroflexi were significantly and negatively correlated with Cr, Zn, Cu, Cd and density and positively correlated with WT (*p* < 0.01). In addition, *Syntrophus* and *Syntrophobacter* were also positively correlated with TN (*p* < 0.05), and *Longilinea* showed a significant positive correlation with SOC and TN (*p* < 0.05). *Escherichia* showed a significant positive correlation with SOC, TN and WT (*p* < 0.01) and a significant negative correlation with Zn, Cd and density (*p* < 0.05), and *Smithella* exhibited a significant negative correlation with Cr, Zn, Cd (*p* < 0.05) and density (*p* < 0.01) and a significant positive correlation with WT (*p* < 0.01). *Candidatus Solibacter* (phylum Acidobacteria) and *Leptolinea* (phylum Acidobacteria) show the opposite trend; i.e., they were positively correlated with Zn, Cu, Cd and density and negatively related to WT, SOC and TN (*p* < 0.05). Additionally, there was a significant positive correlation between *Gemmatimonas* and TK, Zn, Cd and density (*p* < 0.05), whereas there was a significant negative correlation with WT (*p* < 0.001). These results indicated that the soil properties in this analysis had obvious effects on the soil bacterial communities, and SOC, TN, Cd, Cr, Zn, Cu, WT and density played a more important role in shaping the local bacterial community compared with other environmental factors.

To further evaluate the relative influences of environmental factors on the main phyla of soils in turfy swamp, i.e., Proteobacteria and Acidobacteria, multiple linear regression analysis was conducted (Appendix A). The result indicated that only WT and density were significantly correlated with the ratio of Proteobacteria to Acidobacteria. On this basis, we use surface fitting regression method to describe the effect of WT and density on the ratio of Proteobacteria to Acidobacteria, which showed that WT and density can perfectly predict the change of Proteobacteria and Acidobacteria in soil (Table 5 and Figure 8).

## 4. Discussion

Many studies have shown that fuel consumption and vehicle wear release many heavy metals, such as Cr, Cd, Cu, Pb and Zn, into the roadside soil, resulting in a high concentration of heavy metals in the soil near the highway [6,17,18]. Our findings are consistent with their results. However, we found no significant difference in the concentration of Pb between the affected and the control area, which may be related to the banning of leaded gasoline. As expected, road drainage did change some physical and chemical properties of the soil in turfy swamps. Some authors have shown that drainage can lead to loss of soil nutrients, resulting in a decline in soil quality [13,28,29]. At the three sites considered in the current study, drainage ditches with a parallel distance of about 3 m from the highway had been constructed to ensure the safe operation of the highway (Figure 1), which caused a large amount of water to be discharged from the affected area. This is the main reason why WT, SOC and TN values in affected area were significantly lower than those in control area. For soil TP, there was no significant difference between the affected and control area, possibly because phosphate is easily immobilized by the soil and water-soluble phosphorus is scarce. In addition, road drainage is also the main reason for the increase of soil density in affected area, which may be due to the accelerated natural subsidence of the soil by drainage. Therefore, road drainage and pollutant emission are the key factors leading to changes in soil environment in this area.

Human disturbance activities can affect the structure and diversity of soil microbial communities by changing local ecological and environmental factors. For example, Urbanová and Bárta [13] researched the microbial communities of soils from bog, fen and spruce swamp forests and demonstrated that long-term drainage had substantial effects on soil biochemical properties and microbial community composition. Shi et al. [30] found that land subsidence due to underground coal mining could affect soil electrical conductivity and water content, thus altering soil microbial community structure and diversity in sandy areas of Western China, and Guo et al. [31] identified that soil microbial community in mining areas is significantly correlated with Cd, Pb and Zn. In the present study, the sequence data obtained indicated significant differences at the genus level in microbial communities, and the great majority of the affected genera belonged to the phyla Proteobacteria and Acidobacteria. 

Proteobacteria and Acidobacteria were the most dominant phyla in soil of turfy swamps, and their relative abundances were susceptible to external environmental changes. For example, Sun et al. [12] found that the relative abundance of Acidobacteria increased but that of Proteobacteria clearly decreased in drained peatland. Our results were similar in that the relative abundance of Acidobacteria increased and that of Proteobacteria clearly decreased under the dual effects of drainage and heavy metal pollution in turfy swamp. Previous studies have shown that the ratio of Proteobacteria to Acidobacteria can be used as an indicator to reflect changes in soil environmental conditions [11]. In this study, the ratio values in affected and control areas were significantly different and were in the ranges of 1.12–2.05 and 3.03–4.31, respectively, which suggests that soil environmental conditions in affected area of turfy swamps did change. However, our results are inconsistent with the results previously reported by Smit et al. [11] showing that the ratio of Proteobacteria to Acidobacteria varied from 0.14 to 0.46, which may be related to different geographical location and nutritional conditions.

However, at the genus level, *Geobacter* was the most abundant genus in all of the samples, and its relative abundance was significantly lower in the affected area. *Geobacter* is an anaerobic bacterium with extracellular respiration, widely distributed in soil and groundwater sediments, and it plays an important role in anaerobic organic matter degradation and anaerobic methane oxidation [32,33]. Road drainage can improve aeration of the surface soil of turfy wetland and reduce substrate availability, which may be the reason why the relative abundance of *Geobacter* was lower in the affected area than in the control area. The abundance of *Methylobacter* was similarly lower, providing additional support for road drainage increasing methane emission, here by reducing *Methylobacter*, a type I methanotroph that assimilates methane-derived C under aerobic conditions and plays an important role in the process of methane emission [34,35]. The genera *Syntrophus*, *Syntrophobacter* and *Syntrophorhabdus* are Gram-negative, present syntrophism with methanogenic microorganisms and are highly efficient in the degradation of aromatic compounds [36,37]. Their relative abundance is significantly related to heavy metals and soil nutrients, indicating that they are sensitive to road drainage and pollutant discharge affected by highways. In addition, several studies have found that different microorganisms respond differently to the toxicity of heavy metals: those microorganisms that are susceptible to toxins abruptly decrease while resistant microorganisms can adapt to environmental changes. For example, considering environmental changes, *Candidatus Solibacter*, *Gemmatimonas*, *Terrimonas* and *Nitrospira* were found to increase in terms of relative abundance in metal-contaminated soil, while the relative abundance of *Longilinea* decreased [16,38]. Our results were consistent with these observations. However, while few studies have reported that *Leptolinea* and *Smithella* are associated with heavy metals, both were affected in our study, which may be explained by the variations of soil available nutrients and heavy metals.

Based on the above analysis, we conclude that the road drainage and pollutant discharge can alter soil environmental factors and consequently affect the bacterial community structure in soils of turfy swamps. Many studies have indicated that the microbial assemblages are driven by the combined effects of multiple environmental factors, rather than by a single factor. For example, Guo et al. [31] found that the environmental factors SOM, pH, Zn, Cd, Pb and H_2_O had a significant effect on microbial community structure, and Zeng et al. [39] observed that pH, SOM and nutrients were the key factors affecting Actinobacteria and Proteobacteria in the Loess Plateau of China. In our study, RDA and correlation analysis were performed to further explore the response of soil bacterial community to environmental factors, with the results showing that SOC, TN, Cd, Cr, Zn, Cu, WT and density were the main factors affecting microbial abundance and community structure. Among them, WT and density were considered to be the most important because they can predict the change characteristics of the two main phyla (Proteobacteria and Acidobacteria) well. WT is an important factor in the wetland ecosystem, and persistent decrease of the WT caused by drainage can affect the anaerobic condition and change the microbial habitats. In addition, many studies have shown that even a short-term drought can change the microbial community in peatland [40,41]. SOC and TN are the most important nutrient components for organisms because they are involved in a number of cellular processes; many studies have documented that soil microorganisms are significantly correlated with SOC and TN in different ecosystems [42,43].

Cd, Cr, Zn and Cu are also important environmental factors that can affect microbial communities. High concentrations of heavy metals can alter bacterial community structure by many ways, such as protein denaturation, cell membrane destruction and inhibition of cell division or enzyme activity, and different bacteria respond differently to them [44]. It was observed that the genera *Candidatus Solibacter* and *Gemmatimonas* showed significantly positive correlations with Cd, Cr, Zn and Cu. Both of these genera are reported to have the ability to protect themselves from metal toxicity. In contrast, the *Longilinea* is sensitive to heavy metals, and it was found to be negatively correlated with Cd, Cr, Zn and Cu in the present study. In addition, some other genera, such as *Syntrophus*, *Syntrophobacter*, *Leptolinea*, *Smithella* and *Syntrophorhabdus*, were negatively correlated with Cd, Cr, Zn and Cu, indicating that heavy metals can inhibit their growth in soil. However, there is a need for further studies in this area. In addition, it is noteworthy that the above microorganisms are not only affected by Cd, Cr, Zn and Cu, but also significantly influenced by the WT, density, SOC and TN, which means that road drainage and traffic-related metals simultaneously determine soil microbial assemblages.

## 5. Conclusions

This study investigated the impact of highway-related activities on the composition and diversity of bacterial communities in turf swamp soil. The results indicated that road drainage and heavy metal emission are two main driving factors leading to significant changes in soil microbiota at the genus level, and most of the affected genera belonged to Proteobacteria and Acidobacteria. The construction of highways in wetland areas, especially the installation of drainage projects, can lead to severe deterioration and even degradation of wetland ecosystems along the route. Wetlands have important ecological and environmental effects, and wetland conservation is of great importance. Mark studied the environmental changes in the wetland construction area, including bacterial activity, bacterial community structure, and carbon content, and found that significant increases in bacterial activity occurred in wetlands constructed by installing berms across waterways [45]. Xie′s research indicates that the addition of exogenous microbial products to wetlands helps remove contaminants from the soil through bioproduction [46]. Li put forward suggestions and methods for the protection of the wetland water environment in terms of pollution sources, pathways and treatment measures [47]. Along highways in turfy swamps, berms can be installed near the roadbed to prevent wetland water loss. Continuous wetland ecological environment monitoring to obtain effective wetland environmental status and the introduction of exogenous microorganisms to improve the ecological environment can all be effective means of protecting wetlands.

Subsequent statistical analyses further revealed that the main environmental factors determining the soil bacterial community in the turfy swamp were WT, density, SOC, TN, Cd, Cr, Zn and Cu. Moreover, WT and density played a more important role than the other factors, and they can be used to predict the change trend of soil Proteobacteria and Acidobacteria. In addition, WT is also the most important factor affecting the soil bacterial alpha diversity. These findings might have implications for wetland restoration and highway construction scheme selection in turfy swamp areas. However, long-term monitoring should be done to study the effects of continuous drainage and heavy metal accumulation on soil microorganisms and how specific functional microorganisms, such as methanogens, response to complex environmental factors.

## Figures and Tables

**Figure 1 ijerph-17-07822-f001:**
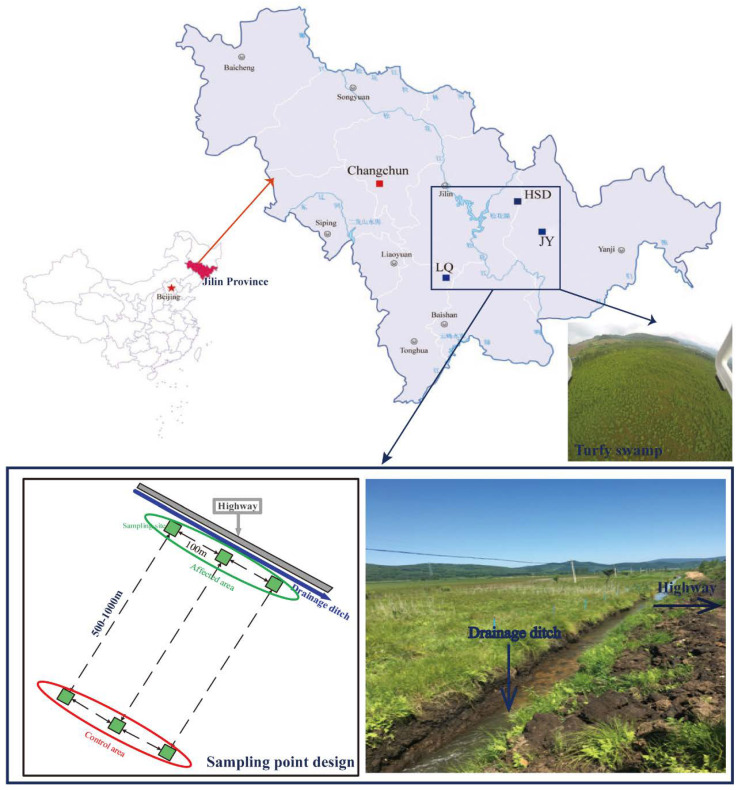
Study area and sampling points design.

**Figure 2 ijerph-17-07822-f002:**
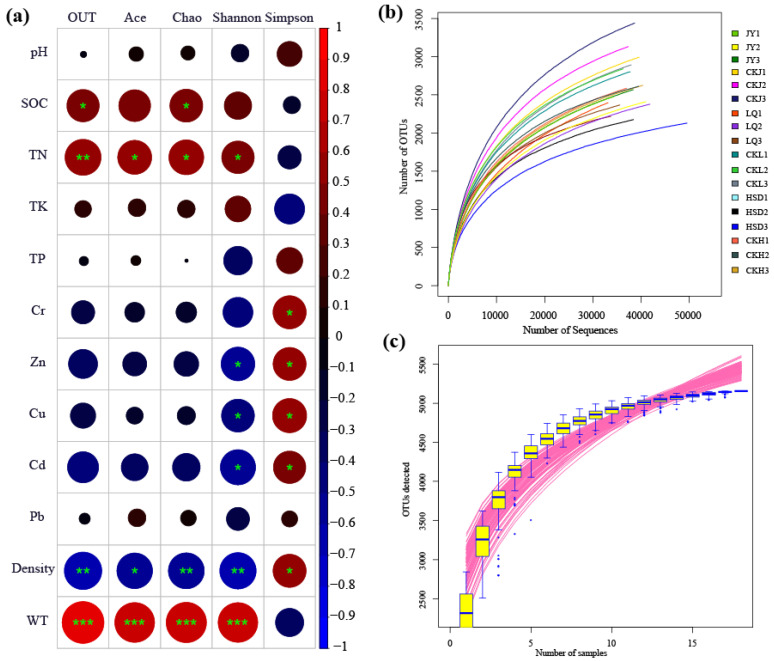
Pearson correlation heat map with correlation coefficient and significance levels based on the bacterial alpha-diversity estimators and environmental factors (**a**). * indicates significance test *p* < 0.05; ** indicates significance test *p* < 0.01; ***indicates significance test *p* < 0.001. Rarefaction curve (**b**) and species accumulation curves (**c**) of the 18 soil samples taken at the three turfy swamp sites.

**Figure 3 ijerph-17-07822-f003:**
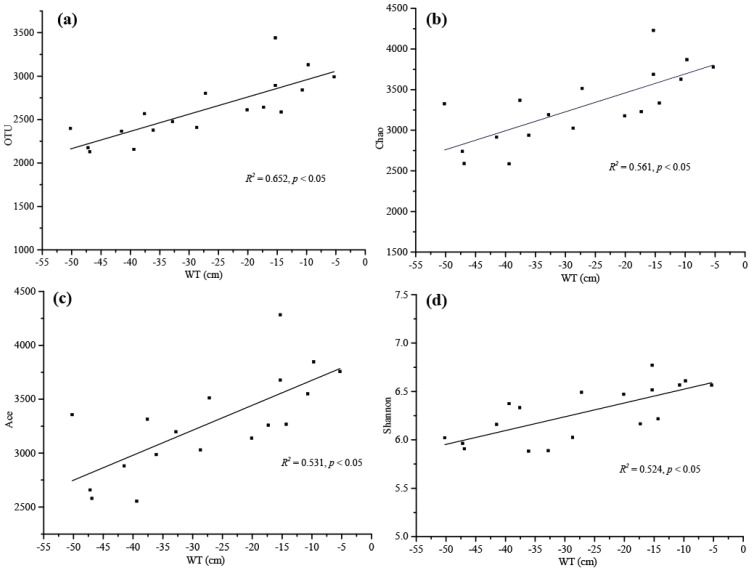
Linear regression relationships between water table (WT) and microbial alpha diversity reported as (**a**) OTU, (**b**) Chao, (**c**) Ace and (**d**) Shannon values.

**Figure 4 ijerph-17-07822-f004:**
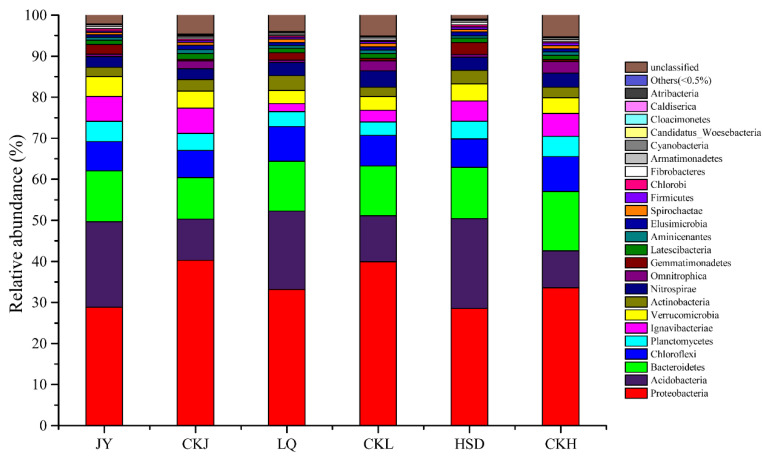
Relative abundance of the bacterial community composition at the phylum level. JY, LQ and HSD represent sampling sites in the highway-affected area, and CKJ, CKL and CKH represent the control sampling sites away from highway in the three turfy swamps.

**Figure 5 ijerph-17-07822-f005:**
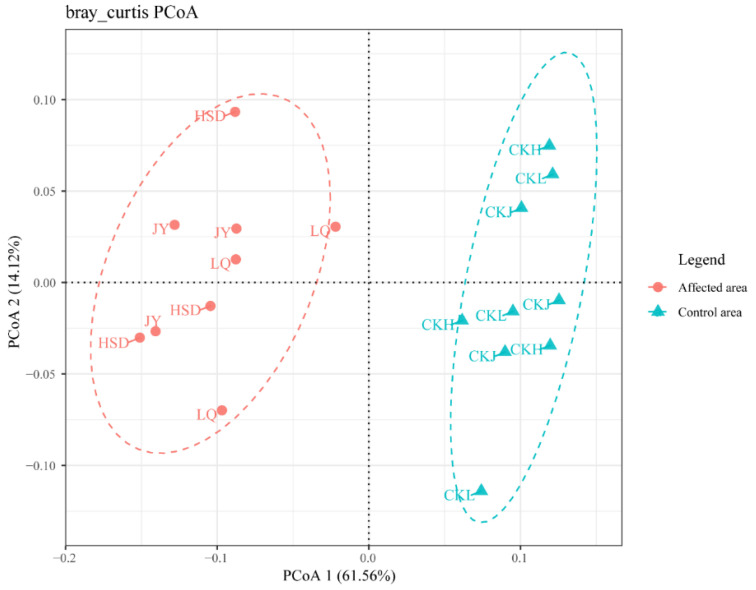
Principal coordinates analysis (PCoA) of the bacterial communities of the three turfy swamps. PCoA 1 and 2 explained 61.56% and 14.12% of the variance, respectively. Points that are closer together on the ordination show communities that are more similar.

**Figure 6 ijerph-17-07822-f006:**
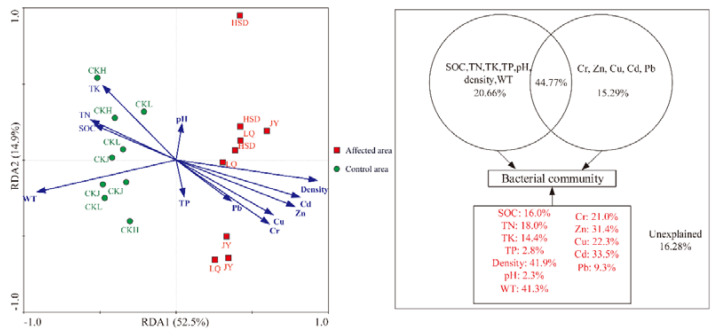
Redundancy analysis (RDA) of the bacterial community compositions and various environmental factors and variation partition analysis of the effects of twelve factors.

**Figure 7 ijerph-17-07822-f007:**
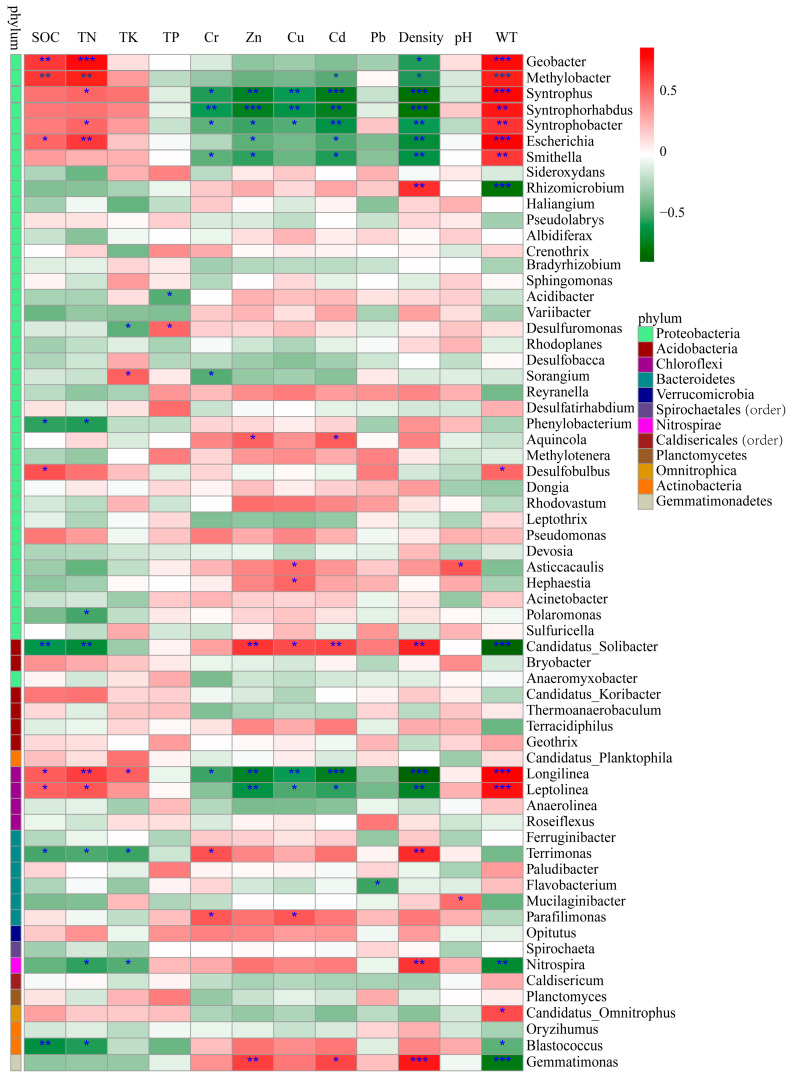
Pearson correlation heat map with correlation coefficient and significance levels based on the bacterial abundance and environmental factors. Colors from green to red represent negative to positive. * indicates significance test *p* < 0.05; ** indicates significance test *p* < 0.01; *** indicates significance test *p* < 0.001.

**Figure 8 ijerph-17-07822-f008:**
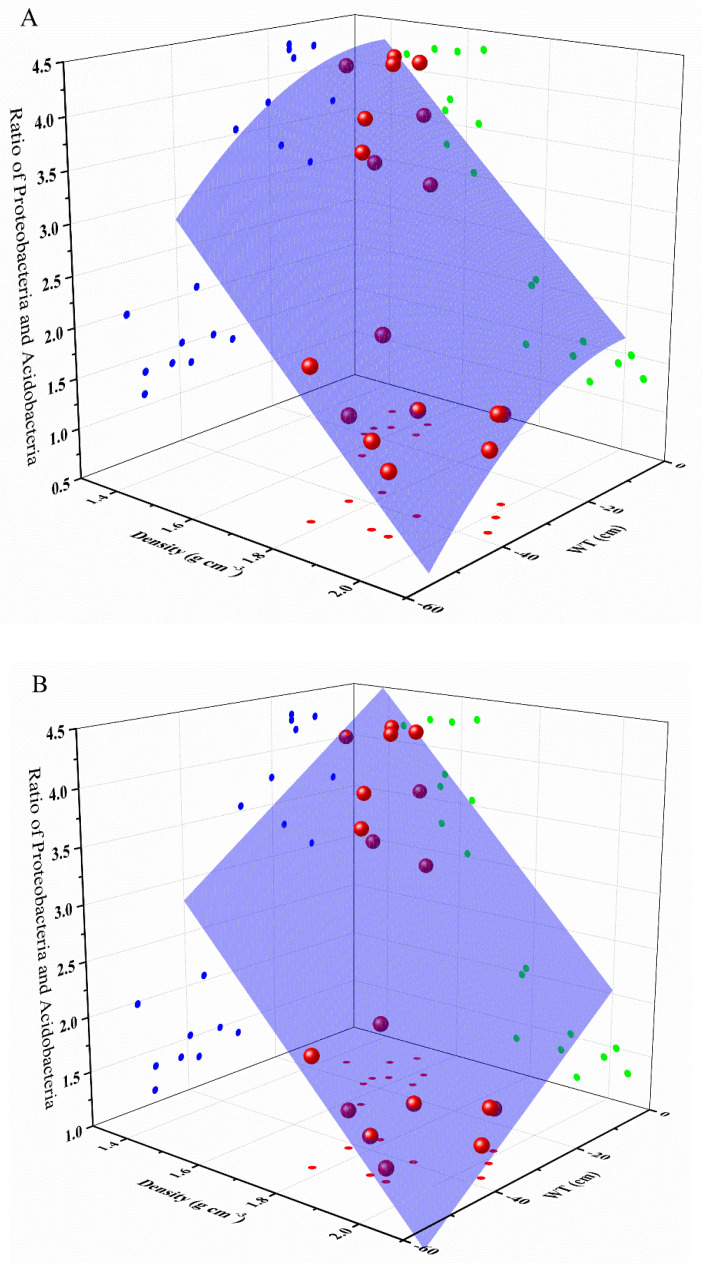
Polynomial model (**A**) and plane model (**B**) for estimating the ratio between Proteobacteria and Acidobacteria with predictor variables of density and WT. The strong positive correlation between the ratio and WT can be seen through the points on the a-plane, while the b-plane shows the unexpected negative correlation between the ratio and density.

**Table 1 ijerph-17-07822-t001:** Physicochemical attributes of soil samples taken from affected and control areas of JY, LQ and HSD sites (*n* = 18).

Parameters	JY		LQ		HSD	
	Affected Area	Control Area	Affected Area	Control Area	Affected Area	Control Area
SOC (g kg^−1^)	130 ± 15.39 ^a^	147 ± 24.26 ^a^	***72.93 ± 8.94 ^b^***	135.67 ± 27.43 ^a^	***101.8 ± 52.99*** ^ab^	132.33 ± 26.31 ^a^
TN (g kg^−1^)	***14.73 ± 2.70 ^b^***	20.17 ± 1.44 ^a^	***8.03 ± 0.56 ^c^***	13.33 ± 0.91 ^b^	***12.19 ± 5.06 ^bc^***	16.70 ± 2.77 ^ab^
TK (g kg^−1^)	***5.61 ± 0.97 ^b^***	6.44 ± 0.72 ^ab^	***7.31 ± 0.69*** ^ab^	8.77 ± 0.12 ^a^	***6.23 ± 1.15*** ^ab^	8.81 ± 3.09 ^a^
TP (g kg^−1^)	1.32 ± 0.09 ^a^	1.28 ± 0.14 ^a^	1.37 ± 0.46 ^a^	1.15 ± 0.07 ^a^	1.22 ± 0.19 ^a^	1.32 ± 0.04 ^a^
Cr (mg kg^−1^)	**121.44 ± 9.36 ^a^**	64.48 ± 10.78 ^b^	**69.05 ± 2.32 ^b^**	46.77 ± 1.09 ^c^	**63.40 ± 5.94 ^b^**	48.01 ± 5.05 ^c^
Zn (mg kg^−1^)	**170.24 ± 27.89 ^a^**	84.14 ± 3.44 ^bc^	**158.43 ± 17.89 ^a^**	69.01 ± 6.28 ^bc^	**93.74 ± 4.95 ^b^**	64.63 ± 4.01 ^c^
Cu (mg kg^−1^)	**51.86 ± 9.91 ^a^**	19.57 ± 1.09 ^b^	**50.34 ± 9.91 ^a^**	18.06 ± 2.76 ^b^	17.02 ± 1.88 ^b^	15.70 ± 0.37 ^b^
Cd (mg kg^−1^)	**0.837 ± 0.11 ^a^**	0.218 ± 0.06 ^cd^	**0.667 ± 0.11 ^b^**	0.152 ± 0.05 ^d^	**0.349 ± 0.08 ^c^**	0.138 ± 0.05 ^d^
Pb (mg kg^−1^)	22.45 ± 1.87 ^a^	19.81 ± 4.43 ^a^	21.55 ± 2.66 ^a^	20.25 ± 2.37 ^a^	19.43 ± 2.71 ^a^	17.56 ± 3.37 ^a^
WT (cm)	***−33.0 ± 4.5 ^b^***	−10.1 ± 5.0 ^a^	***−42.6 ± 13.1 ^bc^***	−17.7 ± 8.5 ^a^	***−44.5 ± 14.4 ^c^***	−17.2 ± 2.9 ^a^
Density (g cm^−3^)	**2.01 ± 0.03 ^a^**	1.54 ± 0.13 ^e^	**1.77 ± 0.01 ^c^**	1.63 ± 0.02 ^d^	**1.89 ± 0.22 ^b^**	1.52 ± 0.06 ^e^
pH	5.81 ± 0.34 ^a^	5.83 ± 0.07 ^a^	5.88 ± 0.39 ^a^	5.54 ± 0.52 ^a^	4.82 ± 0.78 ^a^	5.58 ± 0.48 ^a^

SOC: soil organic carbon; TN: total nitrogen; TK: total potassium; TP: total phosphorus; WT: water table. Different lowercase letters indicate significant differences between different soil samples (*p* < 0.05), with “^a^” indicating the largest group of values in the group, followed by “^b^”, “^c^” and etc. A **bold** value indicates a physicochemical attribute in an affected site that is significantly higher than that in the control site; a ***bold italic*** value indicates a physicochemical attributes in an affected site that is significantly lower than that in the control site (*p* < 0.05).

**Table 2 ijerph-17-07822-t002:** Characteristics of soil bacterial richness and diversity indices in different soil samples.

Sample Code	Reads		0.97					
	Raw Sequences	Trimmed Sequences	Coverage	OTU	Ace	Chao	Shannon	Simpson
JY-A	50,260	40,118	0.98	2483	3180 ± 143 ^c^	3194 ± 171 ^c^	6.28 ± 0.23 ^a^	0.0072 ± 0.0113 ^a^
JY-C	49,895	38,528	0.97	3187	3961 ± 280 ^a^	3958 ± 238 ^a^	6.64 ± 0.31 ^a^	0.0080 ± 0.0011 ^a^
LQ-A	46,772	36,862	0.98	2380	3075 ± 249 ^c^	3059 ± 230 ^c^	6.12 ± 0.24 ^a^	0.0093 ± 0.0017 ^a^
LQ-C	48,862	37,321	0.98	2844	3579 ± 86 ^b^	3608 ± 87 ^b^	6.52 ± 0.04 ^a^	0.0079 ± 0.0013 ^a^
HSD-A	52,611	41,076	0.98	2153	2597 ± 54 ^d^	2637 ± 87 ^d^	6.28 ± 0.16 ^a^	0.0083 ± 0.0029 ^a^
HSD-C	50,237	38,937	0.98	2607	3221 ± 72 ^c^	3246 ± 81 ^c^	6.08 ± 0.35 ^a^	0.0069 ± 0.0014 ^a^

JY: Jiangyuan; LQ: Longquan; HSD: Huangsongdian; A: affected area; C: control area; OTU: the operational taxonomic units. Different lowercase letters indicate significant differences between different soil samples (*p* < 0.05), with “^a^” indicating the largest group of values in the group, followed by “^b^”, “^c^” and etc.

**Table 3 ijerph-17-07822-t003:** Relative abundance of phylum (%) for sequence reads in the three sites (mean ± standard deviation).

	JY Site		LQ Site		HSD Site	
	Affected Area (JY)	Control Area (CKJ)	Affected Area (LQ)	Control Area (CKL)	Affected Area (HSD)	Control Area (CKH)
Proteobacteria	***28.81 ± 1.08 ^c^***	40.28 ± 1.22 ^a^	***33.17 ± 3.45 ^b^***	39.94 ± 3.12 ^a^	***28.53 ± 1.65 ^c^***	33.57 ± 2.90 ^b^
Acidobacteria	**20.85 ± 2.27 ^a^**	10.02 ± 1.11 ^b^	**19.05 ± 3.83** ^a^	11.15 ± 1.37 ^b^	**21.90 ± 4.30 ^a^**	9.02 ± 1.99 ^b^
Bacteroidetes	12.41 ± 1.97 ^a^	10.08 ± 1.76 ^a^	12.14 ± 3.59 ^a^	12.17 ± 4.91 ^a^	12.48 ± 4.20 ^a^	14.44 ± 2.81 ^a^
Chloroflexi	7.08 ± 2.94 ^a^	6.65 ± 1.24 ^a^	8.46 ± 0.72 ^a^	7.43 ± 1.27 ^a^	6.95 ± 2.17 ^a^	8.49 ± 2.05 ^a^
Planctomycetes	4.97 ± 0.20 ^a^	4.12 ± 0.91 ^a^	3.66 ± 1.68 ^a^	3.29 ± 0.50 ^a^	4.30 ± 1.05 ^a^	4.96 ± 1.24 ^a^
Ignavibacteriae	6.05 ± 0.50 ^a^	6.18 ± 1.44 ^a^	1.96 ± 0.67 ^b^	2.83 ± 1.03 ^b^	4.91 ± 1.07 ^a^	5.52 ± 1.33 ^a^
Verrucomicrobia	**4.80 ± 1.20 ^a^**	4.14 ± 1.07 ^ab^	***3.16 ± 0.55 ^b^***	3.30 ± 0.29 ^ab^	4.18 ± 0.27 ^ab^	3.89 ± 1.13 ^ab^
Actinobacteria	2.31 ± 1.94 ^a^	2.85 ± 1.45 ^a^	3.64 ± 1.63 ^a^	2.30 ± 0.48 ^a^	3.27 ± 0.54 ^a^	2.51 ± 0.99 ^a^
Nitrospirae	2.62 ± 1.05 ^a^	2.77 ± 2.23 ^a^	3.22 ± 1.12 ^a^	4.03 ± 2.38 ^a^	3.21 ± 0.41 ^a^	3.44 ± 0.95 ^a^
Omnitrophica	***0.57 ± 0.21 ^c^***	1.97 ± 0.13 ^b^	***0.54 ± 0.22 ^c^***	2.43 ± 0.29 ^ab^	***0.66 ± 0.17 ^c^***	2.82 ± 0.78 ^a^
Gemmatimonadetes	**2.38 ± 0.61 ^ab^**	0.34 ± 0.10 ^c^	**1.83 ± 0.36 ^b^**	0.57 ± 0.19 ^c^	**2.91 ± 0.96 ^a^**	0.51 ± 0.16 ^c^
Latescibacteria	1.04 ± 0.37 ^a^	1.46 ± 0.91 ^a^	1.05 ± 0.26 ^a^	1.22 ± 0.11 ^a^	1.06 ± 0.34 ^a^	1.07 ± 0.21 ^a^
Aminicenantes	0.71 ± 0.20 ^a^	0.96 ± 0.29 ^a^	0.66 ± 0.32 ^a^	0.78 ± 0.48 ^a^	0.59 ± 0.20 ^a^	0.79 ± 0.27 ^a^
Elusimicrobia	0.61 ± 0.33 ^a^	1.02 ± 0.43 ^a^	0.79 ± 0.29 ^a^	0.82 ± 0.18 ^a^	0.90 ± 0.22 ^a^	0.72 ± 0.06 ^a^
Spirochaetae	0.58 ± 0.14 ^a^	0.69 ± 0.04 ^a^	0.79 ± 0.13 ^a^	0.83 ± 0.21 ^a^	0.66 ± 0.27 ^a^	0.83 ± 0.33 ^a^
Firmicutes	0.38 ± 0.12 ^a^	0.45 ± 0.21 ^a^	0.49 ± 0.27 ^a^	0.48 ± 0.26 ^a^	0.62 ± 0.16 ^a^	0.64 ± 0.13 ^a^
Chlorobi	0.53 ± 0.08 ^a^	0.43 ± 0.27 ^a^	0.39 ± 0.14 ^a^	0.30 ± 0.07 ^a^	0.49 ± 0.32 ^a^	0.32 ± 0.16 ^a^
Fibrobacteres	0.47 ± 0.31 ^a^	0.44 ± 0.07 ^a^	0.38 ± 0.12 ^a^	0.48 ± 0.02 ^a^	0.59 ± 0.07 ^a^	0.49 ± 0.13 ^a^
Armatimonadetes	**0.47 ± 0.12 ^ab^**	0.31 ± 0.05 ^b^	**0.36 ± 0.13 ^ab^**	0.31 ± 0.05 ^b^	**0.54 ± 0.14 ^a^**	0.41 ± 0.13 ^ab^
Cyanobacteria	0.13 ± 0.01 ^a^	0.11 ± 0.04 ^a^	0.14 ± 0.09 ^a^	0.13 ± 0.05 ^a^	0.13 ± 0.05 ^a^	0.11 ± 0.04 ^a^
Candidatus_Woesebacteria	0.02 ± 0.02 ^a^	0.03 ± 0.03 ^a^	0.04 ± 0.04 ^a^	0.04 ± 0.02 ^a^	0.01 ± 0.00 ^a^	0.04 ± 0.01 ^a^
Cloacimonetes	0.03 ± 0.01 ^a^	0.02 ± 0.01 ^a^	0.04 ± 0.03 ^a^	0.04 ± 0.02 ^a^	0.03 ± 0.00 ^a^	0.06 ± 0.02 ^a^
Caldiserica	0.02 ± 0.01 ^a^	0.07 ± 0.01 ^a^	0.01 ± 0.00 ^a^	0.01 ± 0.00 ^a^	0.02 ± 0.01 ^a^	0.02 ± 0.02 ^a^
Atribacteria	0.00 ± 0.00 ^a^	0.21 ± 0.05 ^a^	0.00 ± 0.00 ^a^	0.01 ± 0.00 ^a^	0.01 ± 0.00 ^a^	0.00 ± 0.00 ^a^
Others (<0.5%)	0.01 ± 0.00	0.01 ± 0.01	0.01 ± 0.00	0.01 ± 0.00	0.01 ± 0.00	0.01 ± 0.00
unclassified	2.16 ± 1.51	4.58 ± 1.65	3.99 ± 1.87	5.08 ± 2.48	1.01 ± 0.63	5.32 ± 2.39

The value 0.00 in the table means the relative abundance of the bacteria < 0.01%. Different lowercase letters indicate significant differences between different soil samples (*p* < 0.05), with “^a^” indicating the largest group of values in the group, followed by “^b^”, “^c^” and etc. A **bold** values indicates that the abundance of a phylum (%) in an affected site is significantly higher than that in the control site; a ***bold italic*** value indicates that the abundance of a phylum (%) in an affected site is significantly lower than that in the control site (*p* < 0.05).

**Table 4 ijerph-17-07822-t004:** Relative abundance of genus (%) for sequence reads in the three sites. (mean ± standard deviation).

	JY Site		LQ Site		HSD Site	
	Affected Area (JY)	Control Area (CKJ)	Affected Area (LQ)	Control Area (CKL)	Affected Area (HSD)	Control Area (CKH)
*Geobacter*	***2.49 ± 1.59 ^bc^***	7.23 ± 2.36 ^a^	***1.16 ± 0.13 ^c^***	3.97 ± 0.20 ^b^	***1.22 ± 0.45 ^c^***	3.10 ± 0.55 ^bc^
*Syntrophus*	***0.06 ± 0.05 ^c^***	1.67 ± 0.74 ^b^	***0.09 ± 0.06 ^c^***	1.84 ± 0.05 ^b^	**0.06 ± 0.01 ^c^**	2.84 ± 0.70 ^a^
*Escherichia*	***0.44 ± 0.21 ^c^***	3.84 ± 1.92 ^a^	***0.39 ± 0.15 ^c^***	2.07 ± 0.81 ^b^	***0.25 ± 0.11 ^c^***	1.48 ± 0.13 ^b^
*Sideroxydans*	0.15 ± 0.02 ^b^	0.15 ± 0.01 ^b^	0.45 ± 0.31 ^a^	0.42 ± 0.11 ^a^	0.14 ± 0.01 ^b^	0.15 ± 0.02 ^b^
*Rhizomicrobium*	**0.59 ± 0.33 ^b^**	0.09 ± 0.02 ^c^	**0.59 ± 0.35 ^b^**	0.15 ± 0.07 ^bc^	**1.58 ± 0.35 ^a^**	0.15 ± 0.08 ^bc^
*Haliangium*	0.80 ± 0.34 ^a^	0.80 ± 0.18 ^a^	0.48 ± 0.04 ^a^	0.41 ± 0.23 ^a^	0.99 ± 0.86 ^a^	0.83 ± 0.42 ^a^
*Pseudol abrys*	0.39 ± 0.28 ^a^	0.29 ± 0.12 ^a^	0.42 ± 0.09 ^a^	0.42 ± 0.33 ^a^	1.11 ± 0.97 ^a^	0.80 ± 0.62 ^a^
*Albidiferax*	0.63 ± 0.48 ^a^	0.61 ± 0.53 ^a^	0.88 ± 0.22 ^a^	0.91 ± 0.25 ^a^	0.53 ± 0.26 ^a^	0.55 ± 0.27 ^a^
*Crenothrix*	0.65 ± 0.37 ^a^	0.76 ± 0.47 ^a^	0.47 ± 0.31 ^a^	0.43 ± 0.29 ^a^	0.45 ± 0.30 ^a^	0.45 ± 0.12 ^a^
*Syntrophorh ab* *dus*	***0.02 ± 0.01 ^b^***	0.73 ± 0.33 ^a^	***0.02 ± 0.01 ^b^***	0.81 ± 0.35 ^a^	***0.09 ± 0.03 ^b^***	1.03 ± 0.41 ^a^
*Smithella*	***0.00 ± 0.00 ^b^***	0.38 ± 0.14 ^ab^	***0.04 ± 0.03 ^b^***	0.79 ± 0.12 ^a^	***0.00 ± 0.00 ^b^***	1.08 ± 0.64 ^a^
*Syntrophobacter*	***0.02 ± 0.01 ^b^***	0.71 ± 0.36 ^a^	***0.05 ± 0.03 ^b^***	0.52 ± 0.27 ^a^	***0.04 ± 0.03 ^b^***	0.33 ± 0.13 ^ab^
*Bradyrhizobium*	0.19 ± 0.03 ^b^	0.22 ± 0.05 ^b^	**0.32 ± 0.27 ^ab^**	0.25 ± 0.18 ^b^	**0.55 ± 0.16 ^a^**	0.48 ± 0.16 ^ab^
*Sphingomonas*	0.10 ± 0.07 ^a^	0.11 ± 0.05 ^a^	0.33 ± 0.23 ^a^	0.47 ± 0.25 ^a^	0.14 ± 0.07 ^a^	0.14 ± 0.01 ^a^
*Acidibacter*	0.31 ± 0.16 ^a^	0.27 ± 0.19 ^a^	0.43 ± 0.42 ^a^	0.36 ± 0.27 ^a^	0.29 ± 0.13 ^a^	0.25 ± 0.12 ^a^
*Variibacter*	0.40 ± 0.18 ^a^	0.28 ± 0.18 ^a^	0.40 ± 0.35 ^a^	0.22 ± 0.13 ^a^	0.55 ± 0.38 ^a^	0.36 ± 0.30 ^a^
*Methylobacter*	***0.19 ± 0.15 ^b^***	0.58 ± 0.13 ^a^	***0.10 ± 0.02 ^b^***	0.38 ± 0.13 ^ab^	0.14 ± 0.01 ^b^	0.30 ± 0.14 ^b^
*Desulfuromonas*	0.03 ± 0.01 ^a^	0.02 ± 0.02 ^a^	0.02 ± 0.02 ^a^	0.01 ± 0.01 ^a^	0.01 ± 0.01 ^a^	0.03 ± 0.02 ^a^
*Rhodoplanes*	0.20 ± 0.09 ^a^	0.16 ± 0.07 ^a^	0.11 ± 0.06 ^a^	0.14 ± 0.12 ^a^	0.44 ± 0.38 ^a^	0.29 ± 0.26 ^a^
*Desulfobacca*	**0.01 ± 0.00 ^a^**	0.05 ± 0.04 ^ab^	0.03 ± 0.02 ^a^	0.13 ± 0.07 ^a^	0.33 ± 0.05 ^a^	0.27 ± 0.07 ^a^
*Sorangium*	0.04 ± 0.04 ^a^	0.05 ± 0.05 ^a^	0.11 ± 0.04 ^a^	0.11 ± 0.01 ^a^	0.14 ± 0.06 ^a^	0.16 ± 0.10 ^a^
*Reyranella*	0.10 ± 0.03 ^a^	0.06 ± 0.04 ^a^	0.11 ± 0.03 ^a^	0.06 ± 0.04 ^a^	0.10 ± 0.04 ^a^	0.06 ± 0.03 ^a^
*Desulfatirh ab* *dium*	0.03 ± 0.02 ^a^	0.05 ± 0.01 ^a^	0.14 ± 0.04 ^a^	0.16 ± 0.15 ^a^	0.01 ± 0.01 ^a^	0.05 ± 0.04 ^a^
*Phenylobacterium*	0.06 ± 0.02 ^a^	0.03 ± 0.00 ^a^	0.06 ± 0.02 ^a^	0.06 ± 0.01 ^a^	0.09 ± 0.08 ^a^	0.05 ± 0.01 ^a^
*Aquincola*	0.29 ± 0.09 ^a^	0.12 ± 0.06 ^a^	0.14 ± 0.10 ^a^	0.07 ± 0.06 ^a^	0.14 ± 0.07 ^a^	0.08 ± 0.03 ^a^
*Methylotenera*	**0.05 ± 0.04 ^ab^**	0.04 ± 0.04 ^b^	**0.12 ± 0.07 ^a^**	0.02 ± 0.02 ^b^	0.02 ± 0.01 ^b^	0.02 ± 0.02 ^b^
*Desulfobulbus*	0.05 ± 0.01 ^a^	0.05 ± 0.05 ^a^	0.01 ± 0.01 ^a^	0.04 ± 0.04 ^a^	0.01 ± 0.01 ^a^	0.04 ± 0.03 ^a^
*Dongia*	0.03 ± 0.01 ^a^	0.02 ± 0.00 ^a^	0.03 ± 0.02 ^a^	0.01 ± 0.01 ^a^	0.05 ± 0.05 ^a^	0.03 ± 0.02 ^a^
*Rhodovastum*	0.03 ± 0.00 ^ab^	0.03 ± 0.03 ^ab^	**0.07 ± 0.02 ^a^**	0.04 ± 0.02 ^ab^	**0.03 ± 0.01 ^ab^**	0.02 ± 0.01 ^b^
*Leptothrix*	0.01 ± 0.00 ^a^	0.02 ± 0.01 ^a^	0.04 ± 0.05 ^a^	0.06 ± 0.06 ^a^	0.06 ± 0.05 ^a^	0.06 ± 0.04 ^a^
*Pseudomonas*	0.06 ± 0.03 ^a^	0.05 ± 0.03 ^a^	0.04 ± 0.02 ^a^	0.04 ± 0.02 ^a^	0.02 ± 0.02 ^a^	0.03 ± 0.01 ^a^
*Devosia*	0.03 ± 0.02 ^a^	0.02 ± 0.01 ^a^	0.03 ± 0.02 ^a^	0.03 ± 0.03 ^a^	0.06 ± 0.04 ^a^	0.04 ± 0.01 ^a^
*Asticcacaulis*	0.02 ± 0.00 ^a^	0.00 ± 0.00 ^a^	0.02 ± 0.02 ^a^	0.01 ± 0.01 ^a^	0.02 ± 0.02 ^a^	0.01 ± 0.01 ^a^
*Hephaestia*	0.01 ± 0.01 ^a^	0.02 ± 0.02 ^a^	0.05 ± 0.05 ^a^	0.00 ± 0.00 ^a^	0.01 ± 0.01 ^a^	0.00 ± 0.00 ^a^
*Acinetobacter*	0.02 ± 0.01 ^a^	0.02 ± 0.01 ^a^	0.02 ± 0.01 ^a^	0.02 ± 0.02 ^a^	0.01 ± 0.01 ^a^	0.01 ± 0.01 ^a^
*Polaromonas*	0.02 ± 0.01 ^a^	0.01 ± 0.01 ^a^	0.04 ± 0.02 ^a^	0.03 ± 0.01 ^a^	0.02 ± 0.02 ^a^	0.02 ± 0.03 ^a^
*Sulfuricella*	0.01 ± 0.01 ^ab^	0.02 ± 0.01 ^ab^	***0.03 ± 0.04*^ab^**	0.04 ± 0.02 ^a^	0.00 ± 0.00 ^b^	0.00 ± 0.00 ^b^
*Candidatus_Solibacter*	**2.23 ± 0.24 ^a^**	1.17 ± 0.26 ^b^	**2.93 ± 0.52 ^a^**	1.33 ± 0.46 ^b^	**2.66 ± 0.58 ^a^**	1.39 ± 0.47 ^b^
*Bryobacter*	0.76 ± 0.41 ^a^	0.71 ± 0.39 ^a^	0.70 ± 0.38 ^a^	0.63 ± 0.13 ^a^	1.53 ± 0.59 ^a^	1.20 ± 0.82 ^a^
*Anaeromyxobacter*	0.18 ± 0.02 ^a^	0.23 ± 0.14 ^a^	0.35 ± 0.22 ^a^	0.37 ± 0.07 ^a^	0.33 ± 0.16 ^a^	0.27 ± 0.10 ^a^
*Candidatus_Koribacter*	0.14 ± 0.05 ^ab^	0.12 ± 0.05 ^ab^	0.01 ± 0.01 ^b^	0.04 ± 0.02 ^b^	**0.38 ± 0.29 ^a^**	0.20 ± 0.18 ^ab^
*Thermoanaerobaculum*	0.02 ± 0.02 ^a^	0.02 ± 0.02 ^a^	0.08 ± 0.07 ^a^	0.10 ± 0.11 ^a^	0.07 ± 0.01 ^a^	0.11 ± 0.09 ^a^
*Terracidiphilus*	0.03 ± 0.02 ^a^	0.01 ± 0.00 ^a^	0.05 ± 0.04 ^a^	0.02 ± 0.02 ^a^	0.04 ± 0.03 ^a^	0.03 ± 0.02 ^a^
*Geothrix*	0.03 ± 0.01 ^a^	0.06 ± 0.04 ^a^	0.05 ± 0.05 ^a^	0.05 ± 0.03 ^a^	0.01 ± 0.01 ^a^	0.02 ± 0.02 ^a^
*Candidatus_Planktophila*	0.00 ± 0.00 ^a^	0.00 ± 0.00 ^a^	0.00 ± 0.00 ^a^	0.01 ± 0.01 ^a^	0.00 ± 0.00 ^a^	0.01 ± 0.01 ^a^
*Longilinea*	***0.02 ± 0.02 ^b^***	0.99 ± 0.44 ^a^	***0.08 ± 0.05 ^b^***	0.85 ± 0.16 ^a^	***0.02 ± 0.01 ^b^***	1.17 ± 0.15 ^a^
*Leptolinea*	***0.07 ± 0.05 ^b^***	0.32 ± 0.19 ^a^	***0.03 ± 0.01 ^b^***	0.26 ± 0.13 ^ab^	***0.08 ± 0.03 ^b^***	0.32 ± 0.20 ^a^
*Anaerolinea*	0.12 ± 0.08 ^a^	0.15 ± 0.11 ^a^	0.11 ± 0.05 ^a^	0.17 ± 0.13 ^a^	0.20 ± 0.06 ^a^	0.30 ± 0.26 ^a^
*Roseiflexus*	0.02 ± 0.01 ^a^	0.02 ± 0.02 ^a^	0.03 ± 0.00 ^a^	0.02 ± 0.02 ^a^	0.02 ± 0.01 ^a^	0.02 ± 0.02 ^a^
*Ferruginibacter*	0.43 ± 0.04 ^a^	0.45 ± 0.22 ^a^	0.41 ± 0.29 ^a^	0.34 ± 0.19 ^a^	0.49 ± 0.32 ^a^	0.33 ± 0.29 ^a^
*Terrimonas*	**0.75 ± 0.26 ^a^**	0.12 ± 0.14 ^b^	**0.33 ± 0.10 ^ab^**	0.12 ± 0.08 ^b^	**0.75 ± 0.17 ^a^**	0.08 ± 0.01 ^b^
*Paludibacter*	0.10 ± 0.06 ^a^	0.12 ± 0.10 ^a^	0.13 ± 0.20 ^a^	0.14 ± 0.11 ^a^	0.03 ± 0.03 ^a^	0.07 ± 0.04 ^a^
*Flavobacterium*	0.13 ± 0.07 ^a^	0.18 ± 0.08 ^a^	0.09 ± 0.07 ^a^	0.06 ± 0.05 ^a^	0.16 ± 0.07 ^a^	0.22 ± 0.07 ^a^
*Mucilaginibacter*	0.04 ± 0.03 ^a^	0.03 ± 0.01 ^a^	0.08 ± 0.05 ^a^	0.07 ± 0.05 ^a^	0.11 ± 0.06 ^a^	0.07 ± 0.05 ^a^
*Parafilimonas*	**0.03 ± 0.02 ^a^**	0.00 ± 0.00 ^b^	**0.01 ± 0.01 ^ab^**	0.00 ± 0.00 ^b^	0.01 ± 0.01 ^ab^	0.01 ± 0.01 ^ab^
*Opitutus*	3.53 ± 0.94 ^a^	2.94 ± 0.71 ^a^	2.58 ± 0.45 ^a^	2.30 ± 0.65 ^a^	2.74 ± 0.39 ^a^	2.82 ± 0.85 ^a^
*Spirochaeta*	0.57 ± 0.23 ^a^	0.57 ± 0.02 ^a^	0.58 ± 0.21 ^a^	0.54 ± 0.12 ^a^	0.53 ± 0.30 ^a^	0.66 ± 0.56 ^a^
*Nitrospira*	**0.46 ± 0.13 ^ab^**	0.04 ± 0.02 ^b^	0.54 ± 0.12 ^ab^	0.11 ± 0.09 ^b^	**0.74 ± 0.12 ^a^**	0.19 ± 0.13 ^b^
*Caldisericum*	0.00 ± 0.00 ^a^	0.01 ± 0.00 ^a^	0.00 ± 0.00 ^a^	0.02 ± 0.00 ^a^	0.00 ± 0.00 ^a^	0.09 ± 0.07 ^a^
*Planctomyces*	0.07 ± 0.00 ^a^	0.08 ± 0.07 ^a^	0.11 ± 0.06 ^a^	0.12 ± 0.06 ^a^	0.09 ± 0.04 ^a^	0.09 ± 0.02 ^a^
*Candidatus_Omnitrophus*	0.17 ± 0.05 ^a^	0.21 ± 0.07 ^a^	0.12 ± 0.07 ^a^	0.39 ± 0.14 ^a^	0.08 ± 0.06 ^a^	0.33 ± 0.06 ^a^
*Oryzihumus*	0.12 ± 0.04 ^a^	0.08 ± 0.07 ^a^	0.06 ± 0.03 ^a^	0.10 ± 0.06 ^a^	0.46 ± 0.14 ^a^	0.17 ± 0.07 ^a^
*Blastococcus*	0.00 ± 0.00 ^a^	0.00 ± 0.00 ^a^	0.001 ± 0.01 ^a^	0.00 ± 0.00 ^a^	0.01 ± 0.01 ^a^	0.00 ± 0.00 ^a^
*Gemmatimonas*	**0.79 ± 0.58 ^a^**	0.03 ± 0.03 ^b^	**0.71 ± 0.40 ^ab^**	0.05 ± 0.02 ^b^	**1.06 ± 0.61 ^a^**	0.05 ± 0.02 ^b^

The value 0.00 in the table means the relative abundance of the bacteria < 0.01%. Different lowercase letters indicate significant differences between different soil samples (*p* < 0.05), with “^a^” indicating the largest group of values in the group, followed by “^b^”, “^c^” and etc. A **bold** value indicates that the abundance of a genus (%) in an affected site is significantly higher than that in the control site; a ***bold italic*** value indicates that the abundance of a genus (%) in an affected site is significantly lower than that in the control site (*p* < 0.05).

**Table 5 ijerph-17-07822-t005:** Applied regression models for estimate the influence of WT and soil density on the ratio of Proteobacteria to Acidobacteria.

Model	Equation	R2	Pr
Plane model	Ratio = 10.48 − 3.98 × density + 0.03 × WT	0.873	*p* < 0.05
Poly2D model	Ratio = 11.12 − 5.04 × density − 0.01 × WT + 0.23 × density^2^ − 6.84 × WT^2^ + 0.003 × density × WT	0.851	*p* < 0.05

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
