# Peer review of "Soil Bacterial Community Structure in Turfy Swamp and Its Response to Highway Disturbance"

_ijerph, 2020, doi:10.3390/ijerph17217822_

Round 1

Reviewer 1 Report

The manuscript rise an important issue related to the impact of the construction and development of highways in turfy swamp areas.

Authors provided an interesting comparison of changes of physical and chemical soil properties. Besides, they measured and evaluated bacterial composition variation through high-throughput sequencing technology. The objective was to show up the main negative effects of highways implement on turfy swamp ecosystems.

The manuscript is interesting; some adjustments are provided to improve the proposed article:

1- Authors must explain abbreviations the first time they are used: WT.

2- Authors should improve materiel and methods part essentially “ Physico-chemical properties analyses” part by including the main steps of the conducted experiment.

3- Table 1 and 2: Letters indicating differences must be on superscript form.

4- Table 2: Authors should revise the organization of the table

5- Scientific names of bacterial genera and species should be on italic form.

Author Response

Dear professor,

Thank you very much for your review of our manuscript. Your careful reading and critical suggestions will help us to greatly improve the quality and readability of the manuscript. We are pleased to receive your comments and have closely studied and synthesised the information and made substantial changes to the original manuscript. 

Comments and Suggestions for Authors:

The manuscript rise an important issue related to the impact of the construction and development of highways in turfy swamp areas.

Authors provided an interesting comparison of changes of physical and chemical soil properties. Besides, they measured and evaluated bacterial composition variation through high-throughput sequencing technology. The objective was to show up the main negative effects of highways implement on turfy swamp ecosystems.

The manuscript is interesting; some adjustments are provided to improve the proposed article:

  • Authors must explain abbreviations the first time they are used: WT.

Response 1: Thanks for your careful reading and valuable suggestion. WT means water table, and it has been explained at line 17 and 78 in Abstract and Introduction part.

  • Authors should improve materiel and methods part essentially “Physico-chemical properties analyses” part by including the main steps of the conducted experiment.

Response 2: We sincerely appreciate the valuable comments. We did not initially describe the specific steps in the manuscript due to the large number of items and tedious steps involved in the physical and chemical properties of the soil. After receiving your suggestion, we agree that the main steps of the physical and chemical property tests should be described as necessary. The main steps of soil organic carbon (SOC), total nitrogen (TN), total phosphorus (TP), total potassium (TK) were described at line 108-153 in the “Physico-chemical properties analyses” part.

  • Table 1 and 2: Letters indicating differences must be on superscript form.

Response 3: Thanks for your suggestion. This has been fixed in each table of the revised manuscript.

  • Table 2: Authors should revise the organization of the table.

Response 4: Thanks for your valuable suggestion. Table 2 shows summary statistics for six sets of bacterial sequencing data from three regions. The data are arranged horizontally to facilitate the enumeration of the data and the observation of comparisons between affected and control areas. The organization of Table 2 has been adjusted to make it easier to read.

5- Scientific names of bacterial genera and species should be on italic form.

Response 4: Thanks for your careful reading and suggestion. We have changed the form of scientific names of bacterial genera and species to italic throughout the manuscript.

We would like to take this opportunity to thank you again for the reviewing process, which has significantly improved the quality of the work.

Yours Sincerely,

Yan Lv

Reviewer 2 Report

General comment: this work is very comprehensive but its large amounts of data are sometimes hard to understand as presented and so there are some areas that need substantial revision as outlined in the enclosed document. The English is generally very good though I have made some corrections. 

Author Response

Dear professor, 

Thank you very much for your review of our manuscript. Your careful reading and critical suggestions will help us to greatly improve the quality and readability of the manuscript. We are pleased to receive your comments and have closely studied and synthesised the information and made substantial changes to the original manuscript.

Comments and Suggestions for Authors:

General comment: this work is very comprehensive but its large amounts of data are sometimes hard to understand as presented and so there are some areas that need substantial revision as outlined in the enclosed document. The English is generally very good though I have made some corrections. 

Response:

Thank you very much for your very careful reading and for your detailed review suggestions. The suggestions for English writing have been incorporated and corrected in the revised version. The revised version uses the "Track Changes" function in Microsoft Word to facilitate your review. The other issues that have been corrected in the manuscript can be summarized as follows.

  1. “Figure 1: On the map, the box around the region of the study hides the “Y” of “JY”.”

Response: Thanks for your careful reading. This has been fixed in the revised manuscript.

  1. In each table of revised version, letters after data indicating differences have been changed to superscript form.
  2. Lines 148-155, please say what SOC, TN, TK, TP are in the text as I had to constantly refer to the base of the table to work out what they were. Also say what Cr Zn Cu etc are as some people may not know all of the chemical symbols. To save space in Table 1, you can use the chemical symbols as you have done.

Response: Thanks for your valuable suggestion. Soil organic carbon (SOC), total nitrogen (TN), total potassium (TK), total phosphorus (TP) and metal elements chromium, cadmium, copper, zinc, plumbum (Cr, Zn, Cu, Cd, and Pb) have been explained in lines 194-201 in the revised version.

  1. “State that the affected area had significantly LOWER values for SOC, TN, TK as the effect is a lowering of these values. And for the metals, there was an INCREASE in values.”

Response: Thanks for your suggestion. A summary sentence has been added to lines 207-208 of the revised version.

  1. “For Table 1, I suggest that values higher at the affected sites be in bold and those that are lower at the affected sites in bold italic to make the higher and lower significant differences readily recognisable.”

Response: Thanks for your suggestion. We have accepted your suggestion to put the higher values for affected locations in bold and the lower values for affected locations in bold italics, and have followed this through to the Table 3 and 4.

  1. Lines 221-222 in revised manuscript: Where is Fig 1? It is not present in my copy of the paper. Table 2 needs to be wider as many words and values are on two lines. Also explain what OTU is as a footnote.

Response: Thanks for your valuable suggestions. There was a clerical error, “Fig. 1” should be Fig. 2, which has been corrected. The organization of Table 2 has been adjusted to make it easier to read. OTU has been explained as a footnote after Table 2.

  1. The authors do not refer to the results in Fig 2 (b) or Fig 2 (c) anywhere in the text. Fig 2 (b) is very hard to follow.

Response: Thanks for your valuable suggestions. Fig. 2 has been explained and refered on lines 219-222 of the revised version.

  1. “I have a suggestion: Include Table S1 in the main paper and have bold where the affected sites were significantly higher, and bold italic where affected sites were significantly lower so that significant differences are readily recognisable.”

Response: Thanks for your valuable suggestions. We accepted your suggestion to put Table S1 in the main paper, numbered Table 3, and changed the font style to highlight the difference.

  1. “For the differences at the genus level, I think Table S2 is sufficient and it should be in the main paper. I think Figure 5 can be omitted as Figure 5 is much too complex and at times the results were different that shown in Table S2.”

Response: Thanks for your valuable suggestions. Initially, Fig. 5 was made to visualize microbial data,and the discrepancies between the graphs and tables were due to different versions of the microbial sequencing data. We agreed with you that Fig. 5 is difficult for readers to understand, and accepted your suggestion to put Table S2 in the main paper, numbered Table 4, and changed the font style to highlight the difference.

  1. “As with Fig 5, I found it very difficult to follow Fig 8. My first question is why you had one set of colours for each Phylum in Fig 5 and a different set of colours in Fig 8. I must say that I found the colours in Fig 5 easier to follow. While I think Fig 5 is not needed, Fig. 8 is useful if the authors make it easier to follow. Many of the comparisons are between members of Proteobacteria, I recommend putting Proteobacteria that have similar correlations together in Fig 8. For instance have Geobacter first and Methylobacter that has similar correlations with SOC, TN and WT second. Have Synthropus (3rd) followed by Syntrophorhabadus (4th), Syntrophobacter (5th) and you can leave Longilinea where it is as it is in a different Phylum.”

Response: Thanks for your careful reading and valuable suggestions. We agree with you, and your suggestions will greatly enhance the quality and readability of our manuscript. We have replaced Fig. 5 with Table S1 to make it easier for the reader to understand, and then “Fig. 8” is now Fig. 7. Following your suggestion, we have changed the order of the bacterial genera in Fig. 7 so that the Proteobacteria with similar correlations are now together. The description of this figure in the text has also been modified accordingly to help the reader better understand it.

We would like to take this opportunity to thank you again for the reviewing process, which has significantly improved the quality of the work.

Yours Sincerely,

Yan Lv

Reviewer 3 Report

The manuscript entitled: “Soil bacterial community structure in turfy swamp and its response to highway disturbance” presents the consequences of highway construction on the environment, especially on the swamp. The authors examined not only the change in the concentration of heavy metals and changes in the physicochemical properties of the soil near the highway and in the natural surroundings, but also verified the impact of these changes in the presence of microorganisms. I highly recommend this article to publication after minor review. Below, there are a few minor suggestions:

  1. Add dot after Fig in line 88.
  2. It is worth adding one paragraph with proposals on what can be done to preserve the consequences of this statute or advertising to minimize the negative effects.
  3. 23.8% of the literature was published before 2010. Please, update the reference.

Author Response

Dear professor,

Thank you very much for your review of our manuscript. Your careful reading and critical suggestions will help us to greatly improve the quality and readability of the manuscript. We are pleased to receive your comments and have closely studied and synthesised the information and made substantial changes to the original manuscript. 

Comments and Suggestions for Autors:

The manuscript entitled: “Soil bacterial community structure in turfy swamp and its response to highway disturbance” presents the consequences of highway construction on the environment, especially on the swamp. The authors examined not only the change in the concentration of heavy metals and changes in the physicochemical properties of the soil near the highway and in the natural surroundings, but also verified the impact of these changes in the presence of microorganisms. I highly recommend this article to publication after minor review. Below, there are a few minor suggestions:

    1.Add dot after Fig in line 88.

Response 1: Thanks for your careful reading and valuable suggestion. This has been fixed in the revised manuscript.

    2.It is worth adding one paragraph with proposals on what can be done to preserve the consequences of this statute or advertising to minimize the negative effects.

Response 2: We sincerely appreciate the valuable comments. The authors agree with the review in that one paragraph should be added with proposals on what can be done to preserve the consequences of this statute or advertising to minimize the negative effects. As suggested, one paragraph about describing the ecological environment management and protection measures of the wetland has been added with reference to recent studies on wetland construction and protection in lines 468-481. Considering the current situation of trufy swamps in China's Changbai Mountains, we propose to install to install berms near the roadbed to reduce the loss of water from the wetland, and to consider the introduction of exogenous microorganisms to improve the ecological environment.

      3.23.8% of the literature was published before 2010. Please, update the reference.

Response 3: Thanks for your careful reading and valuable suggestion. We take note of the issue and accept your suggestion. Two recent studies on the effects of heavy metal contamination on soil microbial abundance and community structure have been referenced in lines 59-64. Three recent studies on wetland construction and protection are cited in lines 472-478.

We would like to take this opportunity to thank you again for the reviewing process, which has significantly improved the quality of the work.

Yours Sincerely,

Yan Lv

Round 2

Reviewer 2 Report

The revisions make the paper much improved so it acceptable if the following two changes are made:

Line 54 and Line 199 replace plumbum with lead (Pb)

Line 254: phylum level is shown in Fig. 4

Author Response

Dear professor,

Thank you very much for your review of our manuscript. After the review of your first round of comments, we have revised the paper based on your suggestions, which has greatly improved the quality and readability of our paper. We are pleased to receive your further comments and have closely studied and synthesised the information and made substantial changes to the manuscript.

Comments and Suggestions for Authors:

The revisions make the paper much improved so it acceptable if the following two changes are made:

Line 54 and Line 199 replace plumbum with lead (Pb)

Line 254: phylum level is shown in Fig. 4

Response:

Thanks for your careful reading and valuable suggestions.

As you suggest, the word “plumbum” has been corrected as “lead” at line 55 and line 202 of the revised manuscript.

At line 259 of the revised mansucript, the tense of the statement has been changed to present tense. Misspellings of words have also been corrected.

We would like to take this opportunity to thank you again for the reviewing process, which has significantly improved the quality of the work.

Yours Sincerely,

Yan Lv
